# Comparative assessment of macrophage responses and antileishmanial efficacy in dynamic vs. Static culture systems utilizing chitosan-based formulations

**Alaa Riezk**[1,2*], **Alec O'Keeffe**[1], **Katrien Van Bocxlaer**[1,3], **Vanessa Yardley**[1], **Simon L. Croft**[1]

**1** Faculty of Infectious and Tropical Diseases, London School of Hygiene and Tropical Medicine, London, United Kingdom, **2** Centre for Antimicrobial Optimisation, Imperial College London, Hammersmith Hospital, London, United Kingdom, **3** Skin Research Centre, Hull York Medical School, University of York, York, United Kingdom

\* a.riezk@imperial.ac.uk

## Abstract

The discovery of novel anti-leishmanial compounds is essential due to the limitations of current treatments and the lack of new drugs in development. In this study, we employed the Quasi Vivo 900 medium perfusion system (QV900, Kirkstall Ltd, UK) to simulate physiological fluid flow, allowing us to compare macrophage responses and therapeutic outcomes under dynamic versus static conditions. After 24 hours, phagocytosis and macropinocytosis decreased in all cell types under flow conditions compared to static cultures. Under slow ($1.45 \times 10^{-9}$ m/s) and faster ($1.23 \times 10^{-7}$ m/s) flow conditions ((simulating *in vivo* lymphatic flow), phagocytosis decreased by around 42.55% and 56.98% in peritoneal macrophages (PEMs), 42.21% and 56.11% in bone marrow-derived macrophages (BMMs), and 49.75% and 63.32% in THP-1 cells, respectively. Similarly, macropinocytosis decreased by approximately 40.7% and 62.2% in PEMs, 34.8% and 60.9% in BMMs, and 33.3% and 59.3% in THP-1 cell line under this same conditions. In this study, we further assessed the impact of medium perfusion on drug efficacy and macrophage functions using a *Leishmania major* amastigote-macrophage assay. We evaluated the performance of both standard and nanoparticle-based drug formulations within dynamic and static culture systems. After 72 hours of medium perfusion, chitosan solution, blank chitosan-sodium tripolyphosphate (TPP) nanoparticles, and amphotericin B (AmB)-loaded chitosan-TPP nanoparticles exhibited a statistically significant reduction in antileishmanial activity by approximately 30-50% under slow flow conditions and 60-80% under faster flow conditions. In comparison, pure AmB showed a 40% decrease in efficacy at slow flow and a 67% decrease at faster flow, both statistically significant. These results highlighted the importance of considering fluid flow dynamics in *in vitro* studies for a more accurate simulation of *in vivo* conditions, potentially leading to better therapeutic strategies for cutaneous leishmaniasis (CL).

**Data availability statement:** The data underlying the results presented in the study are available in the Supporting Information files, and from https://researchonline.lshtm.ac.uk/id/eprint/4656180/.

**Funding:** Alaa Riezk's doctoral project received funding from the Council for At-Risk Academics (CARA, UK) and the London

School of Hygiene and Tropical Medicine (LSHTM). Alec O'Keeffe received funding from the Biotechnology and Biological Sciences Research Council [grant number BB/M009513/1]. The funders had no role in study design, data collection and analysis, decision to publish, or preparation of the manuscript.

**Competing interests:** The authors have declared that no competing interests exist.

## Introduction

Traditionally, *in vitro* studies of cell behaviour and pathogen interactions have relied on static culture systems where cells are maintained in a stationary medium. While these systems have provided valuable insights, they fail to replicate the dynamic conditions present in living tissues. Static cultures lack the continuous nutrient and waste exchange, mechanical forces, and fluid flow that cells experience *in vivo*, which can significantly influence cellular functions and interactions [1–3]. To address these limitations, dynamic culture systems have been developed. These systems use continuous medium perfusion to create a flow environment that better mimics the physiological conditions encountered by cells in their natural tissue context, thereby enhancing the relevance of experimental models [4,5].

Leishmaniasis, caused by protozoan parasites of the genus *Leishmania*, encompasses a spectrum of diseases with distinct clinical manifestations. These parasites alternate between an extracellular, motile promastigote form within the sandfly vector and an intracellular amastigote form that proliferates within the phagolysosomal compartments of mammalian macrophages [6–8]. The disease manifests primarily as visceral leishmaniasis (VL), which affects the liver and spleen and can be fatal if untreated, and cutaneous leishmaniasis (CL), characterized by self-limiting skin lesions that can cause disfigurement [7,9]. In CL, the skin sites of infection expose macrophages to the dynamic environment of interstitial fluid, where the exact flow rates have not been extensively characterized but are estimated to range from 0.1 to 2 µm/s in uninfected human skin [10–12]. This microenvironment plays a crucial role in the interactions between *Leishmania* parasites and host macrophages. Unlike VL, where infected macrophages experience plasma flow rates, understanding the dynamics of interstitial fluid flow in CL lesions is essential for simulating the complex interactions between parasites and macrophages in the skin [10–12].

To simulate some of the more complex interactions between the parasite and macrophages in the host, we selected the Quasi Vivo 900 medium perfusion system (QV900) with a 6-chamber optical tray. This system enables the culture of cells under continuous medium flow, mimicking interstitial fluid dynamics encountered by macrophages during *Leishmania* infection. By replicating these physiological conditions *in vitro*, we aim to enhance the relevance and reliability of our experimental model compared to traditional static culture systems [12].

The aim of this study was to investigate the effects of dynamic medium perfusion using the Quasi Vivo 900 (QV900) system on cellular responses and the antileishmanial efficacy of chitosan-based formulations against *Leishmania major*. Specifically, we sought to:

- Evaluate how dynamic fluid flow conditions influence macrophage functions such as phagocytosis and micropinocytosis, in response to *Leishmania* infection. Previous research by O'Keeffe *et al* [12] showed that dynamic fluid flow decreased these functions in mouse peritoneal macrophages (PEMs). We extended this investigation to mouse bone marrow monocytes and THP-1 cells to provide a broader understanding of macrophage responses under dynamic conditions.

- Assess the antileishmanial activity of chitosan-based formulations under dynamic culture conditions compared to static cultures.

- Extend the evaluation of antileishmanial drug solutions [13] to drug nanoparticles under dynamic culture conditions. This is of interest as nanoparticles offer enhanced stability, targeted delivery and controlled release properties, potentially leading to improved therapeutic outcomes in cutaneous leishmaniasis.

## Materials and methods

### Medium perfusion system

The Quasi Vivo (QV) medium perfusion systems (Kirkstall Ltd, Rotherham, UK) were chosen for this study due to their capability to allow direct observation of infected cells exposed to various medium perfusion rates and continuous monitoring of infection. Specifically, the QV900, a six-chamber optical tray, permits the connection of chambers in series and provides medium flow speeds at the cell surface that fall within the range of interstitial fluid flow rates observed in humans for a flow rate of 360 μl/min [12,14].

Previous modelling of the QV500 demonstrated that at a flow rate of 360 μl/min, the medium speed at the cell surface mimics human interstitial fluid flow rates. The Quasi Vivo 900 (QV900) chambers are 22 mm deep, which affects the fluid dynamics differently compared to the shallower 12 mm QV500 chambers. To adapt to the increased depth of the QV900, we inserted a 3D-printed Nylon 12 block into some chambers. This adjustment allowed us to better control the effective chamber depth, providing a more accurate simulation of interstitial fluid conditions and ensuring that our findings are relevant and comparable to those from shallower chambers [10–12,15]. Mathematical and computational modelling were employed to calculate the appropriate insert height to ensure that the cell surface flow speeds would match the reported range for interstitial flow in the skin. All six chambers of the QV900 were connected in series, with inserts placed in the last three chambers. A peristaltic pump (Parker Hannifin, UK), located outside the $CO_2$ incubator, continuously circulated the culture medium through the system, maintaining the desired flow conditions. Mean fluid speeds at the macrophage cell surface were estimated to be $1.45 \times 10^{-9}$ m/s and $1.23 \times 10^{-7}$ m/s for cells at the base of the chamber and cells on an insert, respectively [12,14].

### Culture systems

*Leishmania* parasites (*L. major, MHOM/SA/85/JISH118*) amastigotes were isolated from mouse skin lesions and allowed to transform into promastigotes. These promastigotes were maintained in Schneider's insect medium (Sigma Aldrich, UK) supplemented with 10% heat-inactivated foetal calf serum (HiFCS) (Harlan, UK) at 26 ̊C. The parasites were routinely passaged through BALB/c mice (Charles River, UK), and low passage number promastigotes ( <passage number 3) were used for experiments, as infectivity has been shown to decrease with extended cultivation [16,17].

Female BALB/c mice aged 6 to 8 weeks, at 18–20 g, were purchased from Charles River Ltd. (Margate, UK, RRID:IMSR_CRL:547). These mice were kept in controlled rooms with humidity of 55% and temperature of 26 ◦C and fed water and rodent food *ad libitum*. All animal work was carried out under a UK Home Office project licence according to the Animal (Scientific Procedures) Act 1986 and the new European Directive 2010/63/EU. The project licence (70/8427) was reviewed by the LSHTM Animal Welfare and Ethical Review Board prior to submission and consequent approval by the UK Home Office. The mice were humanely killed via $CO_2$ and a secondary method to confirm death. Protocols for the isolation of peritoneal macrophages were approved by the LSHTM Animal Welfare and Ethics Review Board.

### Macrophages

**THP-1 cells.** THP-1 cells (TIB-202™; ATCC, via LGC standards, Teddington, Middlesex, UK) were cultured in RPMI-1640 medium with added L-glutamine and 10% HiFCS. These cells were kept in an incubator set at 37 °C with 5% $CO_2$ and were transferred to fresh medium

once a week at a ratio of 1/10 (cells to fresh medium). For differentiation from monocytes to macrophages, THP-1 cells were treated with 20 ng/mL phorbol 12-myristate 13-acetate (PMA) and incubated for 24 hours at 37 °C with 5% $CO_2$.

**Mouse bone marrow monocytes.** Bone marrow-derived macrophages (BMM) were isolated from the femurs of female BALB/c mice (Charles River Ltd, Saffron Walden, Essex, UK). Using a 25 g needle, the bone cavity was flushed with 5 mL of ice-cold Dulbecco's modified Eagle's medium (DMEM). The collected DMEM was then centrifuged, and the bone marrow progenitor cells were resuspended in RPMI-1640 supplemented with 10% HiFCS and penicillin/streptomycin (pen/strep). The mouse bone marrow monocytes were seeded at a density of 25 million cells per T175 flask. Additional RPMI-1640 with 10% HiFCS and pen/strep, along with macrophage colony-stimulating factor (m-CSF) to achieve a final concentration of 50 ng/mL of m-CSF, was added. The cells were then incubated for 7 days at 37 °C before the mature macrophages were harvested.

**Mouse peritoneal macrophages (PEMs).** Mouse peritoneal macrophages were isolated from female CD1 mice (8-10 weeks of age; IMSR_CRL:022) 24 hours after a peritoneal injection of 0.5 mL of 2% starch solution in sterile phosphate buffered saline. The macrophages were harvested by performing an abdominal lavage using cold RPMI-1640 medium with 1% penicillin and streptomycin. The collected cells were then centrifuged at 500 g for 15 minutes at 4 °C, washed with RPMI-1640 medium, and resuspended in RPMI-1640 medium containing 10% HiFCS [17–19].

## Infection of macrophages with *L. major* promastigotes

Macrophages were seeded onto 12mm round glass coverslips (Bellco, US) in 24-well plates (Corning, UK) at a density of 4 x 10^5 cells per well in RPMI-1640 medium supplemented with 10% HiFCS. After incubating the plates at 37°C with 5% $CO_2$ for 24 hours, stationary phase *L. major* promastigotes were counted and prepared in medium at varying concentrations (ranging from 2 x 10^5 to 6 x 10^7) to achieve initial parasite to macrophage ratios ranging from 0.5:1 to 15:1. The promastigotes were added to the macrophage cultures, and the plates were then placed in a 34°C incubator with 5% $CO_2$ for 24 hours, which is a relevant temperature for cutaneous leishmaniasis.

Following this incubation, two-thirds of the glass coverslips were transferred to a medium perfusion system and maintained under flow conditions at a speed of 360 μl/min for 72 hours in the same 34°C, 5% $CO_2$ environment. The remaining coverslips served as the static control, with macrophages cultured in the same medium without flow. After the incubation period, the cells were fixed with methanol (Sigma, UK) and stained using Giemsa's stain (VWR, UK).

The infection rate of the macrophages was assessed visually using a microscope by counting the number of infected cells per 100 macrophages. The percentage infection values are presented as mean ± standard deviation[17].

## Measurement of macrophage functions

**Phagocytosis.** Phagocytosis of macrophages (PEMs, BMMs, and THP-1) was assessed using 2 μm diameter fluorescent red-labeled latex beads (carboxylate-modified polystyrene; Sigma-Aldrich, UK). Macrophages were infected with *L. major* promastigotes and subjected to the three flow conditions described. After incubation periods ranging from 0.5 to 24 hours, cells were washed with ice-cold PBS to remove non-internalized beads. Cells were then lysed with 0.5% Triton X-100 in 0.2 M NaOH. Phagocytosis was assessed by analyzing cell lysates using a fluorescence plate reader (Spectramax M3) with excitation and emission wavelengths set at 575 nm and 610 nm, respectively. The plate reader was calibrated using standard

solutions containing varying amounts of latex beads mixed with cell lysates. The uptake of latex beads was quantified as the number associated per milligram of cellular protein, with protein content determined using a Micro BCA protein kit (Thermo Fisher, UK) following the manufacturer's protocol. In control experiments, cytochalasin D at a concentration of 1 µg/ml (Sigma-Aldrich, UK) was used to inhibit phagocytosis. Macrophages were pre-incubated with cytochalasin D for 2 hours prior to the addition of latex beads. Phagocytosis was observed to be completely inhibited after 0.5, 1, 2, and 4 hours of cytochalasin D treatment, with 90% inhibition maintained after 24 hours [20,21].

**Macropinocytosis.** Macropinocytosis was measured using pHrodo Red dextran (Thermo Fisher, UK), which fluoresces in an acidic environment. This fluorescent dye exhibits pH-sensitive emission, with fluorescence intensity increasing in acidic environments and minimal emission at neutral pH levels. Following infection with parasites, macrophages were subjected to the previously described three flow conditions. After three washes with Live Cell Imaging Solution (ThermoFisher, UK), the macrophages were cultured in RPMI-1640 supplemented with 10% HiFCS and 40 µg/mL pHrodo Red dextran (1 mL per well) at 34°C with 5% $CO_2$ for durations of 0.5, 1, 2, 4, and 24 hours under each flow condition.

At each time point, the cells were washed again with Live Cell Imaging Solution, and macropinocytosis activity was assessed using a Spectramax M3™ fluorescence plate reader with excitation and emission wavelengths set at 560 nm and 585 nm, respectively. To inhibit macropinocytosis, 10 µg/ml chlorpromazine hydrochloride (Sigma-Aldrich, UK), a known inhibitor, was incubated with the macrophages for 2 hours prior to the addition of the fluorescence-labeled dextran dye. The inhibition efficacy was complete after 0.5, 1, 2, and 4 hours of chlorpromazine hydrochloride treatment, and 90% inhibition was observed after 24 hours [22].

## Drug and nanoparticle preparation and characterisation

All drug formulations used in this study, including amphotericin B (AmB), chitosan solution, blank chitosan-TPP nanoparticles, and AmB-loaded chitosan-TPP nanoparticles, were precisely prepared and characterized as previously described by Riezk et al. [17,18].

Briefly, amphotericin B (Purity ≥ 95%, Cambridge Bioscience, Cambridge, UK) was prepared by dissolving it in DMSO to make a 10 mM stock solution, which was then diluted in RPMI-1640 (Sigma, Gillingham, UK) with 10% HiFCS and referred to as AmB.

For chitosan nanoparticles, 1 g of high molecular weight chitosan (MW = 310–375 kDa, Sigma) was dissolved in 100 mL of 1% acetic acid solution (Sigma) at room temperature, with continuous stirring for 24 hours until a clear solution was achieved. The pH was adjusted to about 6.5 using 2 M sodium hydroxide (NaOH, Sigma) and a pH meter (Orion Model 420 A, Thermo Fisher Scientific, Waltham, MA, USA).

To prepare the nanoparticles, 10 mL of TPP aqueous solution (without AmB for blank nanoparticles, or with 0.5 mL of AmB for drug-loaded particles) was added dropwise to the chitosan solution (10 mL) while stirring for 5 minutes. The suspension was then sonicated for 15 minutes using a probe sonicator (Soniprep 150, Richmond Scientific Ltd., Chorley, UK) to reduce particle size. The suspension was filtered through a 0.2 µm syringe filter (Sigma, UK) to remove aggregates and larger particles. The nanoparticles were purified and concentrated by centrifugation (8000 × g) using 30 kDa centrifugal filters (Spin-X UF concentrators, Corning, Sigma, UK), which removed unentrapped AmB, as only small molecules (MW < 30 kDa) could pass through. The suspension was then lyophilized in a freeze dryer (Micro Modulyo, Richmond Scientific Ltd.) with 5% sucrose as a cryoprotectant. After 48 hours, the lyophilized nanoparticles were collected, weighed, and stored at 4°C.

Encapsulation efficiency (EE) and drug loading per particle mass were calculated using these formulas:

- **EE (%)** = $100 \times$ (weight of total AmB - weight of free AmB)/ weight of total AmB

- **Drug Loading (%)** = $100 \times$ (weight of total AmB - weight of free AmB)/ (weight of chitosan + weight of TPP)

AmB was analysed using a 1260 Infinity Agilent HPLC system. The column and settings used as previously reported in [17,18].

The nanoparticles' physical characteristics, including average diameter, zeta potential, and polydispersity index, were measured using a ZetaSizer (Malvern Instruments Ltd., Malvern, UK) with data analysis performed using Malvern ZetaSizer software v 7.11.

## Impact of medium flow on anti-leishmanial drug and nanoparticle efficacy

PEMs were seeded onto glass coverslips and infected with promastigotes under static conditions for 24 hours. After this period, two-thirds of the coverslips containing infected macrophages were transferred to a medium perfusion system and exposed to flow conditions at a rate of 360 µL/min for 72 hours in a 34°C, 5% $CO_2$ incubator. These coverslips were either placed at the bottom of the well or on top of an insert. The remaining coverslips served as the static control, with macrophages maintained in the same culture medium without flow. This setup created three different culture medium flow rates: static (0 m/s), low flow ($1.45 \times 10^{-9}$ m/s; cells at the bottom of the well), and high flow ($1.23 \times 10^{-7}$ m/s; cells on top of inserts).

Following the initial 24 hours of static infection, the medium used for drug activity studies was supplemented with varying concentrations of drugs. After 72 hours, all coverslips were removed, methanol-fixed, Giemsa-stained, and microscopically examined to count the number of infected macrophages and assess the drug-induced reduction in parasite infection (Fig 1). Data were analyzed using non-linear sigmoidal curve fitting (variable slope) with Prism Software (GraphPad, Surrey, UK).

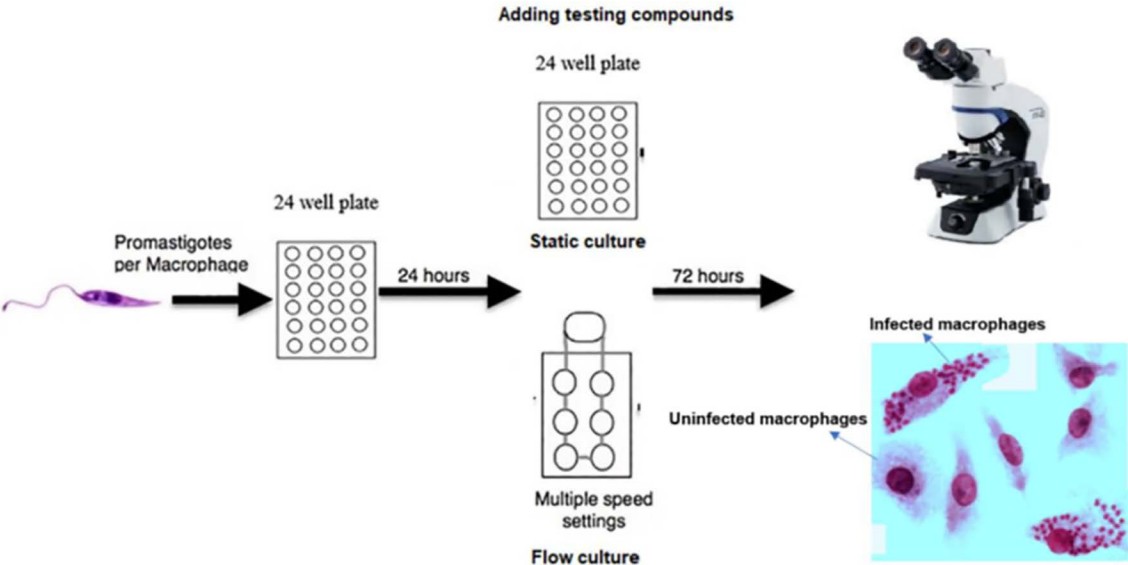

**Fig 1. Schematic overview of evaluation of the anti-leishmanial activity in static and flow culture systems [ 12].**

## Results

### Macrophage functions

**Phagocytosis.** Phagocytosis of latex beads by uninfected and *L. major*-infected macrophages (PEMs, BMMs, or THP-1) showed a clear time-dependent response (Fig 2), with phagocytosis increasing with the duration of incubation. Infected macrophages (infection rate > 80%) exhibited significantly higher phagocytosis compared to uninfected ones after 24 hours under static conditions (Fig 2, $p < 0.05$ by t-test). Specifically, PEMs and BMMs demonstrated significantly higher phagocytosis of latex beads than THP-1 cells (Fig 2, $p < 0.05$ by one-way ANOVA).

The impact of medium perfusion on phagocytosis was also assessed. Flow conditions resulted in a significant reduction in phagocytosis by infected macrophages (Fig 3). After 24 hours of incubation, phagocytosis decreased from $530 \pm 30 \times 10^5$, $519 \pm 30 \times 10^5$, and $398 \pm 22 \times 10^5$ beads/mg protein by PEMs, BMMs, and THP-1 cells, respectively, in static cultures to $304 \pm 32 \times 10^5$, $299.9 \pm 24 \times 10^5$, and $200 \pm 30 \times 10^5$ beads/mg protein by PEMs, BMMs, and THP-1 cells, respectively, under slow flow conditions ($1.45 \times 10^{-9}$ m/s). Under faster flow conditions ($1.23 \times 10^{-7}$ m/s), phagocytosis further decreased to $231 \pm 28 \times 10^5$, $227.6 \pm 25 \times 10^5$, and $144 \pm 18 \times 10^5$ beads/mg protein by PEMs, BMMs, and THP-1 cells, respectively (Fig 3, $p < 0.05$ by one-way ANOVA).

**Macropinocytosis.** Macropinocytosis of pHrodo Red dextran by uninfected and *L. major*-infected macrophages (PEMs, BMMs, or THP-1) showed a clear time-dependent response (Fig 4). Infected macrophages had significantly higher macropinocytosis levels compared to uninfected cells, with PEMs, BMMs, and THP-1 cells showing increases from $19.02 \pm 1.1$, $16.5 \pm 1.1$, and $8 \pm 1.1$ µg/mg protein to $25.3 \pm 0.9$, $23 \pm 0.8$, and $13.5 \pm 0.8$ µg/mg protein, respectively, after 24 hours under static conditions (Fig 4, $p < 0.05$ by t-test).

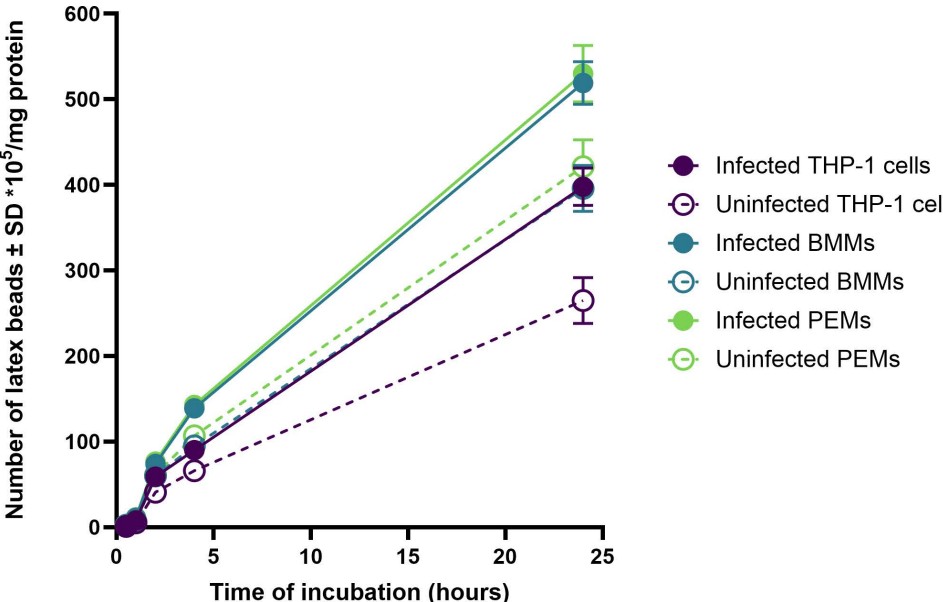

**Fig 2. Phagocytosis of fluorescent latex beads (2 µm) by uninfected and infected PEMs, BMMs, and THP-1 (C) in static culture system.** There is a significant increase in phagocytosis by infected macrophages compared to uninfected ones (p<< 0.05 by t-test). Infection rate was >> 80%. The data show means ±± standard deviations (SD), n == 3. The experiment was reproduced two additional times, confirming similar results. All data are provided in the supporting information (S1, S2 and S3 Tables).

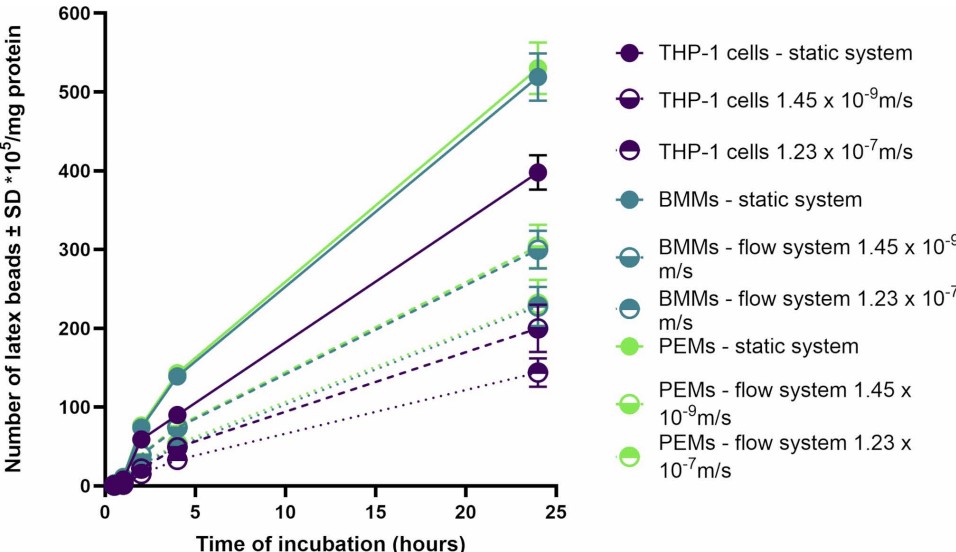

**Fig 3. Phagocytosis of fluorescent latex beads (2 μm) by infected PEMs, BMMs and THP-1 in the three culture systems (static, slow flow rate 1.45 x 10−−9 m/s and fast flow rate 1.23 x 10−−7 m/s).** Phagocytosis is significantly higher in static than in flow system (p< <0.05 by one-way ANOVA). Infection rate was >> 80%. The data show means ±± standard deviations (SD), n == 3. The experiment was reproduced two additional times, confirming similar results. All data are provided in the supporting information (S4, S5 and S6 Tables).

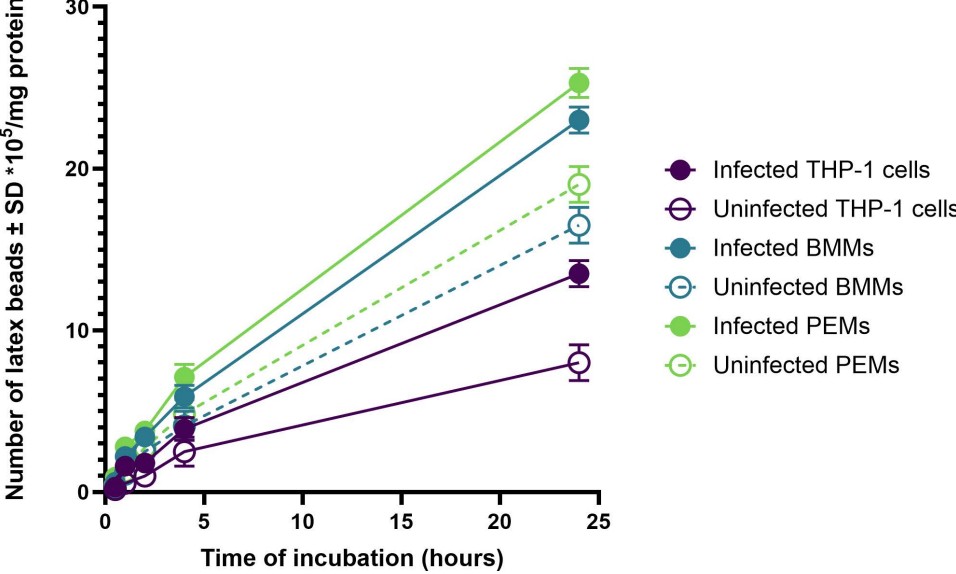

**Fig 4. Macropinocytosis of pHrodo Red dextran by uninfected and infected PEMs, BMMs and THP-1) in static culture system.** There is a significant increase in macropinocytosis by infected PEMs compared to uninfected ones (p< <0.05 by t- test). Infection rate was >> 80%. The data show means ±± standard deviations (SD), n == 3. The experiment was reproduced two additional times, confirming similar results. All data are provided in the supporting information (S7, S8 and S9 Tables).

The effects of medium perfusion on macropinocytosis were also evaluated. Under flow conditions, macropinocytosis was significantly reduced (Fig 5), with higher flow speeds causing greater reductions. After 24 hours of incubation with pHrodo Red dextran,

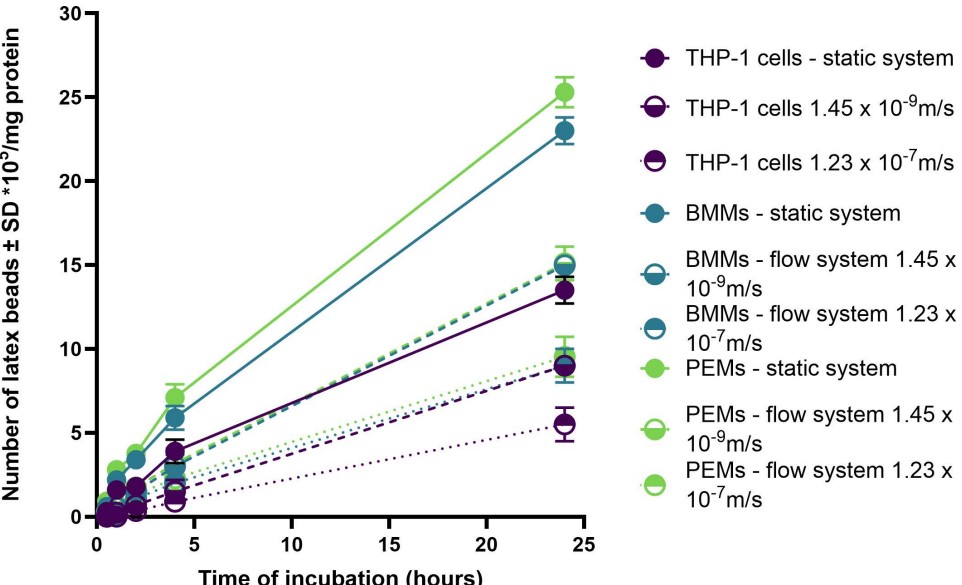

**Fig 5. Macropinocytosis of pHrodo Red dextran by infected PEMs, BMMs and THP-1 at the three culture systems (static, slow flow rate 1.45 x 10−−9 m/s and fast flow rate 1.23 x 10−−7 m/s).** Macropinocytosis is significantly higher in static than in flow systems (p<< 0.05 by one-way ANOVA). Infection rate was >> 80%. The data show means ±± standard deviations (SD), n == 3. The experiment was reproduced two additional times, confirming similar results. All data are provided in the supporting information (S10, S11 and S12 Tables).

macropinocytosis decreased from 25.3 ± 0.9, 23 ± 0.8, and 13.5 ± 0.8 µg/mg protein by PEMs, BMMs, and THP-1 cells, respectively, under static conditions to 15.1 ± 1, 14.99 ± 0.3, and 9 ± 0.3 µg/mg protein by PEMs, BMMs, and THP-1 cells, respectively, under slow flow (1.45 x $10^{-9}$ m/s). The reduction was more pronounced at a higher flow speed (1.23 x $10^{-7}$ m/s), with macropinocytosis levels decreasing to 9.54 ± 1.2, 9 ± 1, and 5.5 ± 1 µg/mg protein by PEMs, BMMs, and THP-1 cells, respectively (Fig 5, p< 0.05 by one-way ANOVA).

## Effects of medium perfusion system on the anti-leishmanial activity of chitosan formulations

The drug encapsulation efficiency (EE) for AmB-loaded chitosan-TPP nanoparticles (AmB-CH-TPP) were both ≥ 90%. The drug loading was approximately 25%. The size of CH-TPP nanoparticles was 48 ± 6 nm before lyophilization and 67 ± 7 nm after lyophilization with 5% sucrose. For AmB-CH-TPP nanoparticles, the size was 57 ± 7 nm before lyophilization and 69 ± 8 nm after lyophilization with 5% sucrose.

A dose-dependent anti-leishmanial activity was observed for all formulations (chitosan solution, blank chitosan-TPP nanoparticles, and AmB-loaded chitosan-TPP nanoparticles) across two medium velocities and static culture (Fig 6). In 72-hour assays, the addition of medium perfusion reduced the anti-leishmanial activity of these chitosan formulations. Chitosan solution, blank chitosan-TPP nanoparticles, and AmB-loaded chitosan-TPP nanoparticles showed significantly higher activity in static culture (0 m/s flow) than in the QV900 system, both at the base of the chamber (1.45 x $10^{-9}$ m/s flow) and on an insert (1.23 x $10^{-7}$ m/s flow) (Table 1 and Fig 6, p< 0.05 by extra sum-of-squares F test). Additionally, a significant difference was observed in EC90 values of pure AmB, with increasing medium perfusion decreasing the efficacy against 90% of amastigotes (Table 1 and Fig 6, p< 0.05).

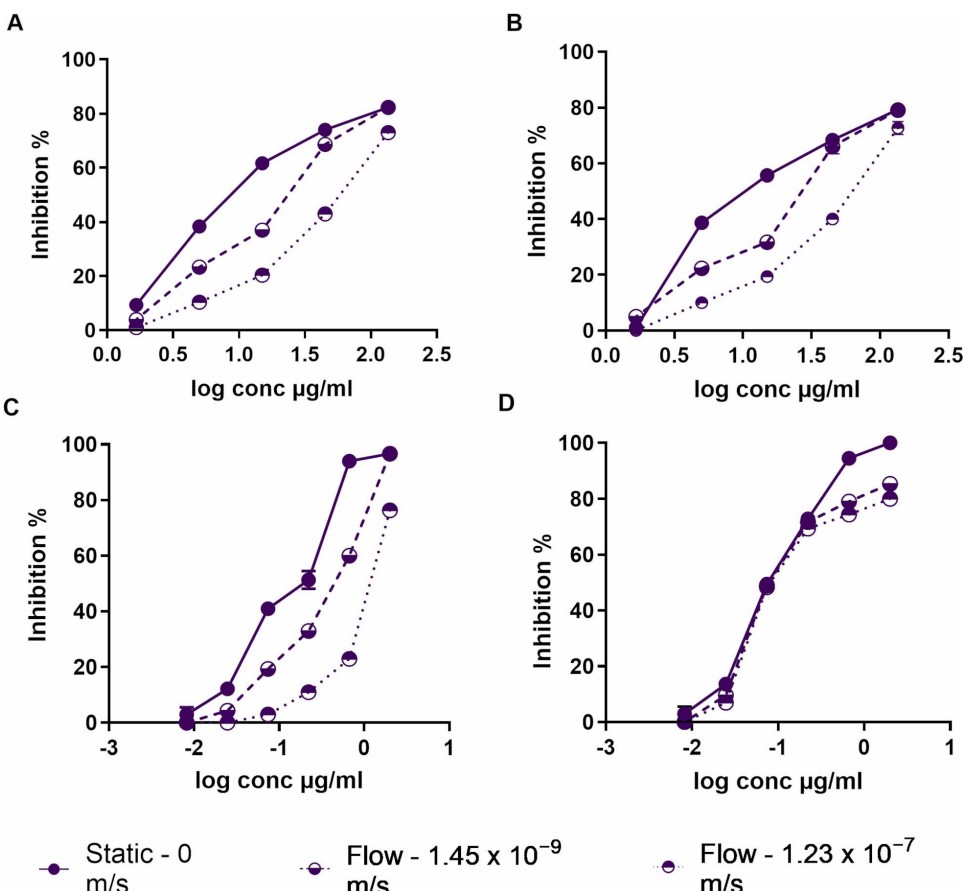

**Fig 6. Dose-response curve of the activity of chitosan solution (A), blank chitosan-TPP nanoparticles (B), AmB loaded chitosan-TPP nanoparticles (C) and AmB solution (pure) (D) against L.** major amastigotes infecting PEMs in pH=6.5 under different flow conditions. Quasi Vivo QV900 system has been used as a flow culture system. Values are expressed as % amastigotes inhibition relative to untreated controls. The data are means ± standard deviations (SD), n = 3. All data are provided in the supporting information (S14 and S15 Tables).

**Table 1. *In vitro* activity of chitosan solution and nanoparticles against *L. major* amastigotes in RPMI-1640 (pH = 6.5) at different flow rates.**

| Compound | Static- 0 m/s | | Flow - $1.45 \times 10^{-9}$ m/s | | Flow - $1.23 \times 10^{-7}$ m/s | |
|---|---|---|---|---|---|---|
| | $EC_{50}$ | $EC_{90}$ | $EC_{50}$ | $EC_{90}$ | $EC_{50}$ | $EC_{90}$ |
| | µg/ml | | | | | |
| Chitosan solution | 10.9 ± 1 | 165 ± 5 | 22.7 ± 1 | 230 ± 15 | 55.3 ± 2 | 455 ± 9 |
| Blank chitosan-TPP nanoparticles | 14.6 ± 4 | 241 ± 26 | 29.3 ± 3 | 299 ± 35 | 53.7 ± 4 | 459 ± 69 |
| AmB loaded chitosan-TPP nanoparticles | 0.1 ± 0.01 | 1 ± 0.1 | 0.4 ± 0.01 | 2.5 ± 0.1 | 1.1 ± 0.02 | 3.5 ± 0.3 |
| AmB solution (Pure) | 0.09 ± 0.01 | 0.5 ± 0.02 | 0.1 ± 0.01 | 0.9 ± 0.1 | 0.1 ± 0.02 | 1.5 ± 0.1 |

*Statistically significant differences were found for the $EC_{50}$ values of chitosan solution, blank chitosan-TPP nanoparticles and AmB loaded chitosan-TPP nanoparticles at static culture (flow of 0 m/s), flow of $1.45 \times 10^{-9}$ m/s and flow of $1.23 \times 10^{-7}$ m/s ($p < 0.05$ by an extra sum-of-squares F test). Initial macrophage infection rate was >80% after 24 h. The data show means ± standard deviations (SD), n = 3. All data are provided in the supporting information (S13 Table).

## Discussion

Our findings revealed a significant increase in cell functions, specifically phagocytosis and macropinocytosis, in *L. major*-infected macrophages (PEMs, BMMs, and THP-1) compared to

uninfected cells. This observation is consistent with previous studies showing that macrophages infected with other *Leishmania* species, such as *L. donovani* or *L. mexicana*, exhibit greater pinocytic rates than their uninfected counterparts, as measured by fluorescent probes[23–25]. Similar results were reported with RAW 264.7 macrophages infected with *L. major*, demonstrating enhanced uptake of fluorescently labelled liposomes[23,24]. These findings suggest that the parasitic infection might induce morphological changes in macrophages or alter their metabolic activity, thereby enhancing their ability to ingest particulate material [26].

Moreover, our study highlighted that PEMs and BMMs exhibited significantly higher phagocytosis and macropinocytosis compared to THP-1 cells. This difference could potentially be attributed to the homogeneous nature and longer lifespan of BMMs and PEMs compared to THP-1 cells [27].

The QV900 culture flow system was employed in this study to address fundamental limitations of static *in vitro* culture systems when investigating cellular responses and the anti-leishmanial activity of compounds and formulations. Static culture systems lack the ability to provide dynamic chemical or physical stimuli to cells, such as concentration gradients, flow, pressure, or mechanical stress caused by fluid movement, all of which are physiologically relevant [12]. In response to concerns regarding potential 'wash out' effects due to flow conditions, previous studies, including those led by O'Keeffe *et al.*, demonstrated that the Quasi Vivo system was optimized to maintain cell viability and *L. major* infection. Specifically, while higher flow rates (360 μl/min) led to reduced infection rates, no complete elimination of macrophages or parasites was observed, suggesting that fluid flow did not significantly interfere with the infection process [12,13].

The effects of medium perfusion rates on host cell phagocytosis and macropinocytosis were also evaluated in this study. We observed that both phagocytosis and macropinocytosis were significantly reduced under flow conditions, with higher medium flow rates leading to further reductions in uptake. This finding aligns with previous reports demonstrating decreased uptake of particles by endothelial cells and macrophages under dynamic flow conditions compared to static cultures [28,29]. For instance, human umbilical vein endothelial cells exposed to shear stress in a dynamic cell culture system showed reduced uptake of FITC-poly(ethylene glycol) diacrylate particles compared to static conditions [28,29]. Similarly, RAW 264.7 macrophages exhibited lower cellular uptake of solid silica particles under dynamic conditions compared to static cultures [29].

One possible explanation for these observations is that static culture conditions may cause sedimentation of particles on the cell surface or lead to localized increases in the concentration of engulfment targets, such as pHrodo Red dextran, thereby enhancing their uptake [30]. In contrast, continuous medium flow in dynamic systems prevents such local accumulation and subsequently reduces particle uptake [31].

We also showed that the medium perfusion system had a significant influence on the anti-leishmanial activity of chitosan solution, blank chitosan-TPP nanoparticles, and AmB-loaded chitosan-TPP nanoparticles. Increasing the flow rates caused a significant decrease in their activity. Similarly, O'Keeffe et al reported that the anti-leishmanial activity of miltefosine and paromomycin against *L. major* amastigotes was reduced under both high and slow flow rates [13].

This decrease in the anti-leishmanial activity of chitosan formulations under the flow system could be attributed to several factors:

i. Waste Product Accumulation: In a static system, waste products from catabolic and xenobiotic metabolism accumulate in the culture medium, causing oxidative stress and leading to the loss of cellular function and viability during the culture time *in vitro*. Dynamic culture conditions can mitigate these issues by distributing nutrients, waste products, and tested substances within the cell culture [32,33].

ii. Drug Sedimentation: Static system conditions can cause the sedimentation of the drug on the cell surface, resulting in a local increase in drug concentrations.Conversely, a flow method for exposing cells to drugs can overcome this problem, leading to a homogeneous dispersion of the drugs and preventing sedimentation [32,33].

iii. Drug Accumulation: The effects of the two medium perfusion conditions used in our study on the accumulation of anti-leishmanial drugs (amphotericin B and miltefosine) have been previously reported by O'Keeffe *et al* [14]. The accumulation of both drugs was significantly higher in the static system compared to the medium perfusion system after 24 hours, and this could be due to a reduction in the rate of drug uptake [14].

The study described here also showed that cell uptake (phagocytosis and micropinocytosis) is reduced significantly by the application of flow compared with static culture conditions. Therefore, this reduction in drug accumulation and macrophage function (phagocytosis and micropinocytosis) are contributing factors to the reduced anti-leishmanial activity seen. While the impact of fluid flow is a key factor influencing macrophage responses, we acknowledge that other factors, such as the proximity to other immune cells and their cytokine interactions, also play crucial roles in shaping macrophage behavior during Leishmania infection. Additionally, the sedimentation pool effect, particularly in lymphoid tissues, can influence phagocytosis and cellular interactions, further adding to the complexity of immune responses. These factors, which are not directly affected by fluid flow, contribute to the overall macrophage response. Therefore, while fluid flow systems are valuable for replicating physiological conditions, macrophage activity is influenced by a combination of factors, and fluid flow alone cannot account for all cellular responses.

In conclusion, our study underscores the critical importance of fluid flow dynamics in evaluating macrophage functions and anti-leishmanial activities. The ability to replicate physiological conditions is valuable, but the impact of fluid flow is the key factor influencing cellular responses. Our findings highlight the necessity of considering flow dynamics in in vitro assessments. Future research should aim to explore how fluid flow affects macrophage behaviour and its implications for understanding disease mechanisms and developing therapeutic strategies.

## Supporting information

**S1 Table. Phagocytosis of fluorescent latex beads (2 μm) by uninfected and infected PEMs, BMMs, and THP-1 in static culture system.** (The data presented in this table were used to generate Fig 2). Values behind the means, standard deviations.
(DOCX)

**S2 Table. Phagocytosis of fluorescent latex beads (2 μm) by uninfected and infected PEMs, BMMs, and THP-1 in static culture system.**
(DOCX)

**S3 Table. Phagocytosis of fluorescent latex beads (2 μm) by uninfected and infected PEMs, BMMs, and THP-1 in static culture system.**
(DOCX)

**S4 Table. Phagocytosis of fluorescent latex beads (2 μm) by infected PEMs, BMMs and THP-1 in the three culture systems (static, slow flow rate 1.45 x 10$^{-9}$ m/s and fast flow rate 1.23 x 10-7 m/s) (The data presented in this table were used to generate Fig 3).** Values behind the means, standard deviations.
(DOCX)

**S5 Table. Phagocytosis of fluorescent latex beads (2 μm) by infected PEMs, BMMs and THP-1 in the three culture systems (static, slow flow rate 1.45 x 10$^{-9}$ m/s and fast flow rate 1.23 x 10-7 m/s).**
(DOCX)

**S6 table. Phagocytosis of fluorescent latex beads (2 μm) by infected PEMs, BMMs and THP-1 in the three culture systems (static, slow flow rate 1.45 x 10$^{-9}$ m/s and fast flow rate 1.23 x 10-7 m/s).**
(DOCX)

**S7 table. Macropinocytosis of pHrodo™ Red dextran by uninfected and infected PEMs, BMMs and THP-1 in static culture system.** (The data presented in this table were used to generate [Fig 4]). Values behind the means, standard deviations.
(DOCX)

**S8 Table. Macropinocytosis of pHrodo™ Red dextran by uninfected and infected PEMs, BMMs and THP-1 in static culture system.**
(DOCX)

**S9 Table. Macropinocytosis of pHrodo™ Red dextran by uninfected and infected PEMs, BMMs and THP-1 in static culture system.**
(DOCX)

**S10 Table. Macropinocytosis of pHrodo™ Red dextran by infected PEMs, BMMs and THP-1 at the three culture systems (static, slow flow rate 1.45 x 10$^{-9}$ m/s and fast flow rate 1.23 x 10$^{-7}$ m/s).** (The data presented in this table were used to generate [Fig 5]). Values behind the means, standard deviations.
(DOCX)

**S11 Table. Macropinocytosis of pHrodo™ Red dextran by infected PEMs, BMMs and THP-1 at the three culture systems (static, slow flow rate 1.45 x 10$^{-9}$ m/s and fast flow rate 1.23 x 10$^{-7}$ m/s).**
(DOCX)

**S12 Table. Macropinocytosis of pHrodo™ Red dextran by infected PEMs, BMMs and THP-1 at the three culture systems (static, slow flow rate 1.45 x 10$^{-9}$ m/s and fast flow rate 1.23 x 10$^{-7}$m/s).**
(DOCX)

**S13 Table. In vitro activity of chitosan solution and nanoparticles against L. major amastigotes in RPMI-1640 (pH = 6.5) at different flow rates.** Values behind the means, standard deviations.
(DOCX)

**S14 Table. Dose-response curve of the activity of chitosan solution (A), blank chitosan-TPP nanoparticles (B) against L.** major amastigotes infecting PEMs in pH = 6.5 under different flow conditions. (The data presented in this table were used to generate [Fig 5], (A) and (B).
(DOCX)

**S15 Table. Dose-response curve of the activity of AmB loaded chitosan-TPP nanoparticles (C) and AmB solution (pure) (D) against L.** major amastigotes infecting PEMs in pH = 6.5 under different flow conditions. (The data presented in this table were used to generate [Fig 5], (C) and (D).
(DOCX)

## Acknowledgments

We are grateful to Dr. Sudaxshina Murdan, UCL School of Pharmacy) for helpful discussions. The authors acknowledge the facilities and the scientific and technical assistance of the LSHTM and UCL School of Pharmacy.

## Author contributions

**Conceptualization:** Alaa Riezk, Vanessa Yardley, Simon L. Croft.

**Formal analysis:** Alaa Riezk.

**Funding acquisition:** Simon L. Croft.

**Investigation:** Alaa Riezk, Alec O'Keeffe, Katrien Van Bocxlaer, Vanessa Yardley, Simon L. Croft.

**Methodology:** Alaa Riezk, Alec O'Keeffe, Katrien Van Bocxlaer, Vanessa Yardley.

**Project administration:** Simon L. Croft.

**Resources:** Simon L. Croft.

**Supervision:** Vanessa Yardley, Simon L. Croft.

**Validation:** Alaa Riezk, Vanessa Yardley.

**Visualization:** Alaa Riezk, Katrien Van Bocxlaer.

**Writing – original draft:** Alaa Riezk, Alec O'Keeffe, Katrien Van Bocxlaer, Vanessa Yardley, Simon L. Croft.

**Writing – review & editing:** Alaa Riezk, Simon L. Croft.

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
