## [Decision Letter · Decision Letter 0]

23 Dec 2024

PONE-D-24-44240Comparative Assessment of Macrophage Responses and Antileishmanial Efficacy in Dynamic vs. Static Culture Systems Utilizing Chitosan-Based FormulationsPLOS ONE

Dear Dr. Riezk,

Thank you for submitting your manuscript to PLOS ONE. After careful consideration, we feel that it has merit but does not fully meet PLOS ONE’s publication criteria as it currently stands. Therefore, we invite you to submit a revised version of the manuscript that addresses the points raised during the review process.

We look forward to receiving your revised manuscript.

Kind regards,

Nisha Singh, Ph.D.

Academic Editor

PLOS ONE

2. To comply with PLOS ONE submissions requirements, in your Methods section, please provide additional information regarding the experiments involving animals and ensure you have included details on (1) methods of sacrifice, and (2) efforts to alleviate suffering.

“Alaa Riezk’s doctoral project received funding from the Council for At-Risk Academics (CARA, UK)  and the London School of Hygiene and Tropical Medicine (LSHTM).  Alec O’Keeffe received funding from the Biotechnology and Biological Sciences Research Council [grant number BB/M009513/1].”

Reviewers' comments:

Reviewer's Responses to Questions

**Comments to the Author**

1. Is the manuscript technically sound, and do the data support the conclusions?

Reviewer #1: Yes

Reviewer #2: Partly

2. Has the statistical analysis been performed appropriately and rigorously? 

Reviewer #1: Yes

Reviewer #2: Yes

3. Have the authors made all data underlying the findings in their manuscript fully available?

Reviewer #1: Yes

Reviewer #2: No

4. Is the manuscript presented in an intelligible fashion and written in standard English?

Reviewer #1: Yes

Reviewer #2: Yes

5. Review Comments to the Author

Reviewer #1: Selective drugs for treatment of leishmaniasis is pentavalent antimony compounds but the emerging parasite resistance beside several side effects are the limiting factors. So, the discovery of novel anti-leishmanial compounds is essential due to the limitations of current treatments and the lack of new drugs in development. This manuscript highlighted the importance of considering fluid flow dynamics in in vitro studies for a more accurate simulation of in vivo conditions. The contents and the findings of the manuscript are good presented and suitable for publication in the journal. And indeed, potentially can lead to better therapeutic strategies for CL.

Reviewer #2: The submitted paper presents the use of a medium perfusion system, that pretends simulate the interstitial flow, in the skin to evaluate macrophage functions and antileishmanial activity of some compounds/drugs.

The authors report an increase in macrophage pinocytosis and bead phagocytosis due to leishmania infection, which supports the use of chitosan nanoparticles formulations for anti-leishmanial activity. Did the authors use other methods to evaluate phagocytosis, as microscopy-based counting (phagocytic index)?

All figures, for differ assays, represent the result of only one experiment in triplicate. The authors should present, or make available, other results to show reproducibility of their assays.

Figure 3 has too much overlapping, making it difficult to see difference in pinocytosis due to flow rate.

The authors attribute several factors to the lower activity of macrophages and antileishmanial activity under the flow system, mostly accumulation on the static system. But do they consider the flow conditions a “wash out” problem? The authors conclude that “The ability to replicate physiological conditions is valuable, but the impact of fluid flow is the key factor influencing cellular responses”. That is too bold. There are more “key factors” related to macrophage cellular responses to leishmania that probably are not affected by fluid flow as the proximity to other immune cell and their cytokines.

Finally, It is important consider the sedimentation pool effect that occurs in lymphoid tissues as important for phagocytosis and cellular interactions.

6. PLOS authors have the option to publish the peer review history of their article (what does this mean?). If published, this will include your full peer review and any attached files.

Reviewer #1: No

Reviewer #2: No

---

## [Author Response · Author response to Decision Letter 1]

17 Jan 2025

2. To comply with PLOS ONE submissions requirements, in your Methods section, please provide additional information regarding the experiments involving animals and ensure you have included details on (1) methods of sacrifice, and (2) efforts to alleviate suffering. Amended

“Alaa Riezk’s doctoral project received funding from the Council for At-Risk Academics (CARA, UK) and the London School of Hygiene and Tropical Medicine (LSHTM). Alec O’Keeffe received funding from the Biotechnology and Biological Sciences Research Council [grant number BB/M009513/1].”

Please state what role the funders took in the study. If the funders had no role, please state: "The funders had no role in study design, data collection and analysis, decision to publish, or preparation of the manuscript." Amended

4. We note that you have included the phrase “data not shown” in your manuscript. Unfortunately, this does not meet our data sharing requirements. PLOS does not permit references to inaccessible data. We require that authors provide all relevant data within the paper, Supporting Information files, or in an acceptable, public repository. Please add a citation to support this phrase or upload the data that corresponds with these findings to a stable repository (such as Figshare or Dryad) and provide and URLs, DOIs, or accession numbers that may be used to access these data. Or, if the data are not a core part of the research being presented in your study, we ask that you remove the phrase that refers to these data. Amended

5. Please review your reference list to ensure that it is complete and correct. If you have cited papers that have been retracted, please include the rationale for doing so in the manuscript text, or remove these references and replace them with relevant current references. Any changes to the reference list should be mentioned in the rebuttal letter that accompanies your revised manuscript. If you need to cite a retracted article, indicate the article’s retracted status in the References list and also include a citation and full reference for the retraction notice. Amended

Reviewers' comments:

Reviewer's Responses to Questions

Comments to the Author

1. Is the manuscript technically sound, and do the data support the conclusions?

Reviewer #1: Yes

Reviewer #2: Partly

2. Has the statistical analysis been performed appropriately and rigorously?

Reviewer #1: Yes

Reviewer #2: Yes

3. Have the authors made all data underlying the findings in their manuscript fully available?

Reviewer #1: Yes

Reviewer #2: No

4. Is the manuscript presented in an intelligible fashion and written in standard English?

Reviewer #1: Yes

Reviewer #2: Yes

5. Review Comments to the Author

Reviewer #1: Selective drugs for treatment of leishmaniasis is pentavalent antimony compounds but the emerging parasite resistance beside several side effects are the limiting factors. So, the discovery of novel anti-leishmanial compounds is essential due to the limitations of current treatments and the lack of new drugs in development. This manuscript highlighted the importance of considering fluid flow dynamics in in vitro studies for a more accurate simulation of in vivo conditions. The contents and the findings of the manuscript are good presented and suitable for publication in the journal. And indeed, potentially can lead to better therapeutic strategies for CL.

Response:

We greatly appreciate your positive feedback and recognition of the importance of our work in simulating in vivo conditions using fluid flow dynamics. Thank you for acknowledging the significance of our findings and their potential to enhance therapeutic strategies for cutaneous leishmaniasis (CL).

Reviewer #2: The submitted paper presents the use of a medium perfusion system, that pretends simulate the interstitial flow, in the skin to evaluate macrophage functions and antileishmanial activity of some compounds/drugs.

The authors report an increase in macrophage pinocytosis and bead phagocytosis due to leishmania infection, which supports the use of chitosan nanoparticles formulations for anti-leishmanial activity. Did the authors use other methods to evaluate phagocytosis, as microscopy-based counting (phagocytic index)?

Response:

Thank you for your thoughtful question. In our study, we measured phagocytosis using fluorescent red-labelled latex beads and a fluorescence plate reader. Macrophages were infected with L. major promastigotes, and bead uptake was quantified by analysing cell lysates with fluorescence. The method was calibrated with standards and expressed as the number of beads per milligram of cellular protein.

Why We Used This Method:

1. Quantitative and Reproducible: This approach provided precise and consistent data, allowing for easy comparison between experimental conditions.

2. Efficient for Large-Scale Experiments: It is faster and more scalable than microscopy, making it ideal for our study with multiple flow conditions.

3. Validated Results: We confirmed the specificity of phagocytosis by using cytochalasin D to block bead uptake, ensuring the accuracy of our method.

Why Not Microscopy-Based Counting:

Microscopy-based counting (phagocytic index) is useful but time-consuming and less practical for large-scale experiments. Our fluorescence-based method was more efficient and provided reliable data.

All figures, for differ assays, represent the result of only one experiment in triplicate. The authors should present, or make available, other results to show reproducibility of their assays.

Response:

In response to your concern regarding the reproducibility of our assays, we will provide additional data in the form of tables as supplementary material for all the experiments reported in the manuscript. These tables will include results from independent replicates, ensuring that the reproducibility of our findings is clearly demonstrated.

Figure 3 has too much overlapping, making it difficult to see difference in pinocytosis due to flow rate.

Response:

We acknowledge that the overlapping in Figure 3 may make it challenging to clearly distinguish the differences in pinocytosis under varying flow rates. Unfortunately, due to the nature of the data, we are unable to separate the figure into different images without losing important context. However, to address this concern, we will provide the raw data and additional detailed tables of the results as supplementary material in the revised manuscript.

This supplementary data will allow for better interpretation and a clearer understanding of the findings.

The authors attribute several factors to the lower activity of macrophages and antileishmanial activity under the flow system, mostly accumulation on the static system. But do they consider the flow conditions a “wash out” problem?

Response:

Response: Flow Conditions and 'Wash Out' Problem: Regarding the concern about flow conditions possibly leading to a 'wash out' effect, we agree that this is a valid point to consider. However, our previous work, including studies led by O’Keeffe et al., demonstrated that the Quasi Vivo system was optimized to maintain cell viability and infection with L. major. In our study, we found that the flow rate itself did not result in complete 'wash out' of the macrophages or parasites. Specifically, higher flow rates (360 μl/min) showed reduced infection rates, but we did not observe a complete elimination of the parasites or macrophages, suggesting that fluid flow did not significantly interfere with the infection process. Moreover, we established a reproducible infection model in this system, which provides a reliable platform for evaluating macrophage responses and drug efficacy under controlled flow conditions.

This explanation has now been added to the manuscript to address the concern regarding potential 'wash out' effects due to flow conditions.

The authors conclude that “The ability to replicate physiological conditions is valuable, but the impact of fluid flow is the key factor influencing cellular responses”. That is too bold. There are more “key factors” related to macrophage cellular responses to leishmania that probably are not affected by fluid flow as the proximity to other immune cell and their cytokines.

Response:

Impact of Fluid Flow on Macrophage Responses: We appreciate your comment regarding the statement on fluid flow being the "key factor influencing cellular responses." We agree that other factors, such as the proximity to other immune cells and their cytokines, are also critical in macrophage responses to Leishmania infection. We have revised the manuscript to acknowledge these additional factors and emphasize that while fluid flow plays a significant role, it is not the sole determinant of macrophage activity. We aim to present a balanced view that considers the complexity of immune responses in the context of flow conditions.

Finally, It is important consider the sedimentation pool effect that occurs in lymphoid tissues as important for phagocytosis and cellular interactions.

Response:

While our study focused on the impact of fluid flow within the in vitro perfusion system, we recognize that such sedimentation effects may be relevant in vivo, particularly in tissues where fluid flow dynamics are more complex.

6. PLOS authors have the option to publish the peer review history of their article (what does this mean?). If published, this will include your full peer review and any attached files.

Do you want your identity to be public for this peer review? For information about this choice, including consent withdrawal, please see our Privacy Policy.

Reviewer #1: No

Reviewer #2: No

---

## [Decision Letter · Decision Letter 1]

5 Feb 2025

Comparative assessment of macrophage responses and antileishmanial efficacy in dynamic vs. static culture systems utilizing chitosan-based formulations

PONE-D-24-44240R1

Dear Dr. Riezk,

We’re pleased to inform you that your manuscript has been judged scientifically suitable for publication and will be formally accepted for publication once it meets all outstanding technical requirements.

Kind regards,

Nisha Singh

Academic Editor

PLOS ONE

Additional Editor Comments (optional):

Reviewers' comments:

Reviewer's Responses to Questions

**Comments to the Author**

1. If the authors have adequately addressed your comments raised in a previous round of review and you feel that this manuscript is now acceptable for publication, you may indicate that here to bypass the “Comments to the Author” section, enter your conflict of interest statement in the “Confidential to Editor” section, and submit your "Accept" recommendation.

Reviewer #2: All comments have been addressed

2. Is the manuscript technically sound, and do the data support the conclusions?

Reviewer #2: Yes

3. Has the statistical analysis been performed appropriately and rigorously? 

Reviewer #2: Yes

4. Have the authors made all data underlying the findings in their manuscript fully available?

Reviewer #2: Yes

5. Is the manuscript presented in an intelligible fashion and written in standard English?

Reviewer #2: Yes

6. Review Comments to the Author

Reviewer #2: Commentaries about the manuscript paper by reviewers were well addressed by the authors and adjustments were made to improve the presentation y accessibility to al relatedn not shown, data.

7. PLOS authors have the option to publish the peer review history of their article (what does this mean?). If published, this will include your full peer review and any attached files.

Reviewer #2: No

---

## [Editor Report · Acceptance letter]

PONE-D-24-44240R1

PLOS ONE

Dear Dr. Riezk,

I'm pleased to inform you that your manuscript has been deemed suitable for publication in PLOS ONE. Congratulations! Your manuscript is now being handed over to our production team.

Kind regards,

on behalf of

Dr. Nisha Singh

Academic Editor

PLOS ONE